# Towards a Synthesis of the Non-Genetic and Genetic Views of Cancer in Understanding Pancreatic Ductal Adenocarcinoma Initiation and Prevention

**DOI:** 10.3390/cancers15072159

**Published:** 2023-04-05

**Authors:** Vishaka Gopalan, Sridhar Hannenhalli

**Affiliations:** Cancer Data Science Laboratory, National Cancer Institute, National Institutes of Health, Bethesda, MD 20814, USA

**Keywords:** cellular heterogeneity, transcriptional heterogeneity, regulatory networks, single cell omics, therapy resistance

## Abstract

**Simple Summary:**

Cancer initiation and progression has been studied in purely genetic terms for decades. Implicit in this view is that all cells in a tissue are phenotypically identical until a single cell acquires a set of mutations that will eventually lead to malignancy. This model does not fully account for numerous clinical and epidemiological findings about cancer incidence and progression. Here, we summarize recent mathematical and biological insights that demonstrate how cells in a tissue can switch between dramatically different phenotypes independent of mutations. We explore how these insights, combined with our detailed understanding of oncogenic mutations, may answer key unexplained aspects of pancreatic ductal adenocarcinoma initiation. Importantly, such a combined model allows for a more nuanced understanding of pre-malignancy, and points the way towards early detection and intervention approaches in high-risk patients.

**Abstract:**

While much of the research in oncogenesis and cancer therapy has focused on mutations in key cancer driver genes, more recent work suggests a complementary non-genetic paradigm. This paradigm focuses on how transcriptional and phenotypic heterogeneity, even in clonally derived cells, can create sub-populations associated with oncogenesis, metastasis, and therapy resistance. We discuss this complementary paradigm in the context of pancreatic ductal adenocarcinoma. A better understanding of cellular transcriptional heterogeneity and its association with oncogenesis can lead to more effective therapies that prevent tumor initiation and slow progression.

## 1. Introduction

### 1.1. Genetic Origins of Cancer

The accumulation of mutations in the cells of origin of a tumor is a key step in the initiation and maintenance of oncogenesis [1,2]. This is certainly true of pancreatic ductal adenocarcinoma (PDAC), where a great majority of PDAC tumors contain mutations in the *KRAS* gene. Mouse models where *KRAS*^G12D^ is induced in the pancreas grow tumors that mimic key clinical aspects of human PDAC tumor progression [3]. Intraductal papillary mucinous neoplasms (IPMNs) and pancreatic intraepithelial neoplasms (PanINs), which are common precursor lesions to PDAC in humans, harbor *KRAS*^G12D^ mutations in nearly 40–60% (IPMN) and 90% (PanIN) of cases [4,5]. While the cell-of-origin in PDAC is debated, PDACs can be derived from both pancreatic acinar cells (which make up nearly 90% of the pancreas) and ductal cells in mouse models [6].

### 1.2. Genetics Does Not Completely Explain Cancer Initiation

This purely genetic view of PDAC initiation implicitly assumes that every pancreatic epithelial cell is equally likely to initiate PDAC until one of them acquires the requisite oncogenic mutations. The role of factors such as diet, smoking and family history in increasing PDAC risk [7] are thought to ultimately change mutation rates in epithelial cells. This is thought to occur either directly through germline mutations or indirectly via tissue inflammation that causes increased epithelial cell division. This line of thinking cannot easily explain certain clinical observations, such as the fact that pancreatic main-duct IPMNs are more likely to lead to PDAC than branch-duct IPMNs [5], that PDAC arises more often in the head of the pancreas than the tail [8], or that a single bout of pancreatitis elevates the risk of PDAC for up to ten years after recovery [9]. These observations may be understood via notions of phenotypic heterogeneity across clonally derived cells. Decades of theoretical and experimental work have demonstrated that clonally derived cells exhibit phenotypic heterogeneity and where the heredity of a cellular state can be encoded by gene regulatory network dynamics [10]. In accordance, it has been shown that precursor cells are more susceptible to specific driver mutations than differentiated cells, and such oncogenic competence is correlated with the chromatin state of the developmental program [11]. Observations of pancreatic epithelial cells during injury and homeostasis, for instance, suggest that different sub-populations of epithelial cells divide during our lifetime [8]. These observations and theories suggest that thinking about cancer initiation through the lens of cellular states (quantified via transcriptomics or other methods) may provide a fruitful unification of the purely genetic and the purely systems-biology views of cancer, and lead to newer therapeutic avenues.

### 1.3. Stochastic Gene Expression and Regulatory Networks Result in Transcriptional Heterogeneity within a Cell Type

A regulatory network refers to the set of interactions, transcriptional or otherwise, between all the genes in a cell. Regulatory networks influence the transcriptional state of a cell, which can be defined as the global gene expression profile of the cell. Early mathematical models [12,13] of gene regulation suggested that each cell type in a multicellular organism can be thought of as a stable transcriptional state. Stability implies that a cell is impervious to many perturbations, i.e., minor changes in its microenvironment, or stochastic changes within a cell, do not dramatically alter its transcriptomic identity or phenotype. The observed statistical clustering of cells by type in single-cell RNA-seq atlases from multiple species support this notion, though statistically distinguishing between different states of a given cell type on the one hand, and different cell types on the other, can be a challenge in such datasets [14]. A cell of a given type is “attracted” towards one of its stable states when it is perturbed by either minor changes in its microenvironment [15] or the fluctuating expression of genes [16] within the cell. Accordingly, at a given point in time, all cells of a given type exhibit transcriptional heterogeneity. Some cell types are more “plastic” than others, i.e., they are able to switch between dramatically different transcriptional states, often under the influence of external signals. The amount of transcriptomic (and ultimately, phenotypic) variation across cells of a given type is a trait under the control of natural selection, with low variation (and robustness to changes in external stimuli) being desirable in some contexts, such as embryonic development [17], and high variation in others, such as immune responses [18]. This pervasive transcriptional heterogeneity has implications not only in development and homeostasis, but also in cancer.

### 1.4. Transcriptional Heterogeneity in Cancer

Cancer is characterized by major transcriptional/epigenetic changes not only in cells of origin but also in other cell types that jointly constitute the tumor microenvironment (TME). The cancer phenotype is ultimately a result of dynamically changing interactions between various cell types in dynamically varying states. Much of the research in oncogenesis, metastasis, drug resistance and relapse has focused on protein-coding mutations. However, a growing body of literature clearly points to an alternate mechanism centered around transcriptional heterogeneity and cellular plasticity that sets the stage for oncogenesis and plays critical roles in tumor progression, metastasis, and therapeutic response. As an example, an early study of normal human mammary epithelial cell culture showed that these cells interconverted between two transcriptionally and phenotypically different states marked by the expression of CD44. CD44^hi^ cells were more stem-like, and additionally, oncogenically transformed CD44^hi^ cells formed tumors in mice more rapidly than transformed CD44^low^ cells [19]. The presence of such populations is dependent on cell culture conditions [20]. Sorted CD44^low^ cell populations diversified over time and switched to a CD44^hi^ state. Furthermore, mammary epithelial cells that detach from the matrix in 3D cultures exhibit gene expression profiles similar to those in pre-malignant breast lesions [21]. As yet another example, using a mouse model of tumor progression from pre-neoplastic hyperplasia to lung adenocarcinoma, Marjanovic et al. identified a high plasticity cell state (HPCS) in pre-malignant *TIGIT-positive* lung lesions exhibiting high growth and differentiation potential towards a malignant state [22]. HPCS in human lung adenocarcinoma tumors is associated with poor survival and greater resistance to chemotherapy in preclinical studies. Non-genetic variation in expression also plays a key role in therapy resistance, with vemurafenib resistance in melanoma being a prominent example. A fraction of cells in clonally derived vemurafenib-sensitive cell lines can stochastically express *AXL* and resist vemurafenib treatment [23] without acquiring a mutation. Thus, non-genetic heterogeneity can prime epithelial cells for oncogenic transformation and some cancer cells to escape therapy. It is likely that this phenotype can be fixed in a cell population through epigenetic or genetic alterations.

## 2. Transcriptional Heterogeneity and Pancreatic Ductal Adenocarcinoma Initiation

### 2.1. Acinar Heterogeneity in Pancreas Homeostasis

An early single-cell RNA-seq study of 108 pancreatic acinar cells from mice [24] detected a sub-population marked by a high expression of STMN1 (a protein that regulates microtubule assembly [25]) that constituted 1% of acinar cells but expanded to 30% during response to pancreatic injury. Another single-cell RNA-seq study of the human pancreas [26] found an acinar sub-population expressing the REG3A protein, a secreted bacterial C-type lectin, which is up-regulated in pancreatitis patients [27] and is involved in the transdifferentiation of acinar cells to ductal cells. A more detailed single-nucleus RNA-seq study of the human pancreas [28] confirmed the presence of *REG3A*-expressing acinar cells in the homeostatic pancreas. 

The presence of cellular heterogeneity among acinar cells, and specifically, the presence of *REG3A*-expressing cells in the homeostatic pancreas, raises the question of whether there exists an acinar sub-population that has higher oncogenic potential. Our group developed a set of statistical tests to detect such “edge cells” [29] in an extensive re-analysis of all available pancreas and PDAC single-cell RNA-seq data [30]. We found that acinar cells, but not ductal cells, from histologically normal pancreas tissues possessed an edge sub-population. Edge cells were statistically farther (in a transcriptomic space) from the average acinar cell, while transcriptionally “drifting” towards a PDAC state. We found that these edge cells up-regulated STMN1 in addition to the transcription factors (TFs) *SOX9* and *PTF1A*, which mark multipotent progenitors in embryonic pancreatic tissue [31]. Remarkably, we also found edge cells in varying proportions across different healthy human pancreases from single-cell RNA-seq datasets, with a strong positive correlation between the fraction of edge-like cells and the donor’s age. Crucially, through somatic variant calling, we ruled out the possibility that edge-like cells were clonally derived. A later study [32] found that inducing *KRAS*^G12D^ expression in mouse pancreatic acinar cells exhibiting a high level of telomerase (Tert^hi^ acinar cells) generated larger numbers of acinar progeny than Tert^low^ acinar cells, implying that Tert^hi^ cells better fit in an oncogenic context. However, as single-cell RNA-seq was not carried out in this study, it is unclear if Tert^hi^ cells are identical to the edge sub-population we detected.

### 2.2. Alternative Non-Genetic Paradigm to Oncogenesis

Our finding that the fraction of edge-like acinar cells in the pancreas increases with age while not being descended from a single ancestor cell may present an alternative to the prevalent paradigm of oncogenesis. Transcriptional changes induced by oncogenic mutations are believed to be a key step in cancer initiation. In tissues such as the skin, colon, and esophagus epithelium [33,34,35] stem-cell clones (i.e., stem cells and their progeny) accumulate mutations over time, with some mutations being oncogenic in nature. However, the existence of progenitor or stem cells within the adult pancreas is controversial. Current lineage-tracing studies [24,32] show acinar cells to be non-cycling during homeostasis until injury induction. Recovery from injury involves acinar cell plasticity, where there is a large but transient increase in the fraction of cycling acinar cells, eventually leading to a homeostatic state devoid of cycling acinar cells. This raises the question of the sufficiency of oncogenic mutations for PDAC initiation. An early mouse model showed that inducing *KRAS*^G12V^ expression in adult acinar cells did not lead to pre-malignant lesions in the pancreas [36], while *KRAS*^G12D^ expression in acinar cells resulted in PanIN formation in adult mice [37]. Remarkably, a more recent study found that when *Kras*^G12D^ was expressed in a small fraction of acinar cells in mice, they were eliminated by their wild-type counterparts [38]. This is believed to be due to cell competition [39], a phenomenon seen in development where “fitter” epithelial cells can induce apoptosis in their less-fit neighbors. Thus, the presence of a *KRAS*^G12D^ mutation in a cell does not necessarily lead to oncogenic transformation. This is because the survival of a *KRAS*^G12D^ cell requires it to be fitter than its wild-type counterparts, which is in turn likely to depend on the tissue microenvironment. Tissue inflammation during pancreatitis resolution cooperates with *KRAS*^G12D^-induced transcriptional changes, resulting in PDAC [40]. The inflammation likely provides the microenvironment in which *KRAS*^G12D^ cells outcompete *KRAS*^WT^ cells and proceed towards malignancy. Our edge-cell signature was up-regulated across all bulk RNA-seq samples of whole pancreatic extracts in mice recovering from pancreatitis induced by caerulein administration, as well as in single-nucleus RNA-seq data of acinar cells in chronic pancreatitis patients [30]. The magnitude of up-regulation was more pronounced in mice bearing *KRAS*^G12D^ mutations than in *KRAS*^WT^ mice. This link between inflammation and cancer initiation is in line with chronic pancreatitis being a risk factor for PDAC and may be interpreted within the paradigm of a tumor as a wound that does not heal [41].

### 2.3. Aging Microenvironment, Edge Cells, Increased Oncogenesis

In human pancreatic islet cells, aging is accompanied by phenotype drift and overall increase in inter-cell transcriptomic variation [42] as the accumulated mutations match signatures of aging. Similar observations have been made across organs in the Tabula Muris Senis cohort [43] of single-cell RNA-seq data across organs over the mouse lifespan. Cells from older mice are easier to oncogenically transform [44], as the deteriorating microenvironment’s aging can promote oncogenesis either via mechanisms related to senescence or stiffening of the extracellular matrix [45,46]. Our analysis finds an accumulation of edge-like cells with ageing but does not provide any obvious explanation. Future work will have to tease apart the contribution of microenvironmental changes with age and other cell-autonomous factors, such as changes in mitochondrial function, DNA methylation and other epigenetic marks underlying this trend. 

### 2.4. Tissue-Specific Oncogenic Effects and Links between KRAS^G12D^ Mutation and Edge State

The question of why the *KRAS*^G12D^ mutation is a common driver of PDAC is unclear. *KRAS*^G12D^ mutations are common only in lung, pancreatic and colorectal adenocarcinomas and are thus not as much of a pan-cancer driver when compared to *TP53* loss. As a rule, most driver mutations tend to be tissue-specific [47] and the reasons for this are not established. Our analysis reveals an insight into why *KRAS*^G12D^ in particular may drive PDAC. We discovered [30] that transcriptomic differences between edge and non-edge acinar cells in mice were highly correlated to transcriptomic differences between acinar cells from *KRAS*^G12D^ and *KRAS*^WT^ mice (Figure 1C). This suggests that the edge state is transcriptionally very similar to the *KRAS*^G12D^-driven state, which may mean that the edge state is susceptible to oncogenic transformation in a similar manner to *KRAS*^G12D^ induction. It is also possible that the *KRAS*^G12D^ mutation may either increase the rate of transition from the non-edge to the edge state or increase the stability of the edge state. Further experiments and analyses are required to determine whether *KRAS*^G12D^ is the only mutation that has this effect, or whether others that commonly occur in pre-malignant pancreatic lesions, such as *GNAS* [4], have similar effects.

As mentioned above, traits that lead to high gene expression heterogeneity can undergo natural selection if it confers survival benefits to an organism. Given the relationships between the edge state, pancreatitis resolution, and PDAC, it is worth speculating on whether the edge state represents a mal-adaptation or an adaptation. On the one hand, aging is associated with an increase in transcriptional heterogeneity [42,43], suggesting increasing dysregulation over this process. This is consistent with the notion that natural selection is less effective in purging mutations, whose effect is apparent late in life [48]. On the other hand, if cells in the edge state are primed to respond to pancreatic injury, then selection might maintain the ability of acinar cells to switch to an edge state to allow for quicker injury resolution. When the pancreas is challenged with injury after recovering from an earlier one, the second injury causes far less damage [49]. Interestingly, this study found that this priming for injury response is coupled with a higher susceptibility to KRAS-driven oncogenesis than cells from an uninjured pancreas. A similar observation has been made in mouse skin, where successive injuries are responded to much more quickly than the first injury [50], although the oncogenic potential of these epithelial cells was not evaluated.

## 3. Perspective and Future Directions

The transcriptional state of a cell is a consequence of regulatory interactions between genes as well as cell-intrinsic and cell-extrinsic stochastic fluctuations in gene expression [51,52], both of which can be influenced by genetics as well as epigenetics. Given that the acinar edge cells, to the best of our knowledge, are not induced by specific mutations, the heritability of the edge state needs to be better investigated, i.e., if an edge acinar cell were to divide, would its daughter cells also stay in an edge state? Over how many cell divisions does the edge state persist? If the edge state is a non-dividing state, how long does an edge state persist before transitioning to a non-edge state? Epigenetic memory can be maintained over tens of cell divisions [53,54,55]. Is that sufficient to affect a fate decision? Carefully designed lineage tracing experiments combined with single-cell multi-omic profiling are needed to definitively answer these questions. 

Early detection is a challenge in PDAC treatment, and indeed in most cancers. Early detection is particularly challenging in the case of PDAC given the relatively low incidence rate of the disease [9]. Given that 90% of PDAC cases are sporadic and do not involve familial history or inherited genetic disease [56], defining a high-risk cohort for screening is a challenge (Figure 1D). However, given that the edge state potentially sits at the cross-roads of pancreatitis and PDAC, the genes up-regulated within the edge state may serve as potential biomarkers of PDAC. The ability of computational methods to impute gene expression within the pancreas and other tissues based on RNA-seq and epigenetic data from blood withdrawn [57] might provide non-invasively accessible biomarkers related to the edge state.

The development of an early detection test is likely to lead to a decrease in PDAC-related mortality but does not immediately suggest a therapy. Aside from the development of an early detection test, ongoing works point to possible early interventions amongst patients with a high risk of PDAC. In the case of colorectal cancer, conditions such as ulcerative colitis and Crohn’s disease are known to increase cancer risk [58,59]. Patients who managed these conditions through the administration of anti-TNF antibodies had a lower risk of colorectal cancer [60,61]. This perhaps suggests a general principle that reversing the state of inflammation within a tissue may reduce cancer risk, which may also partly explain the lower risk of lung, colorectal and bladder cancer in smokers who have reduced cigarette usage [62,63,64]. The prioritization of drugs that can effectively revert a pathological transcriptomic state has led to repurposed drugs for cancer and other diseases [65]. Drugs that reverse the edge signature may thus represent candidates that lower PDAC risk in high-risk cohorts. Future research on effective ways to reverse cellular states presents a promising complementary avenue to combat cancer-related mortality.

## Figures and Tables

**Figure 1 cancers-15-02159-f001:**
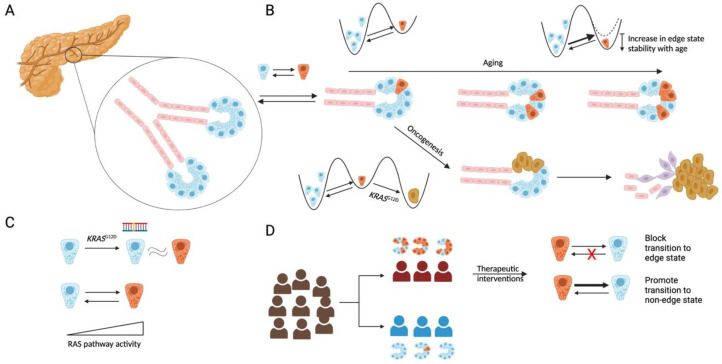
(**A**) Schematic of the human pancreas, with the exocrine compartment consisting of acinar cells (blue) and ductal cells (red). (**B**) Acinar cells can switch to an edge state (brown) characterized by low expression of acinar identity genes. The fraction of edge cells increases with age, which may be due to an increased transition rate to the edge state or an increase in the stability of the edge state, or a combination thereof. Cells in the edge state are predisposed to malignant transformation with the acquisition of a *KRAS^G12D^* mutation, leading to malignancy, with malignant cells existing in different phenotypic states (purple or brown) (**C**) Edge cells have high Ras activity, and the edge-like state phenocopies (at a transcriptomic level) the *KRAS^G12D^* mutation in mice. (**D**) Acinar cells likely exist in an edge state in patients that are at a high risk for PDAC. Therapeutic interventions to block transitions to an edge state, or promote transitions away from the edge state, may lower PDAC risk.

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
