# Peer review of "Towards a Synthesis of the Non-Genetic and Genetic Views of Cancer in Understanding Pancreatic Ductal Adenocarcinoma Initiation and Prevention"

_cancers, 2023, doi:10.3390/cancers15072159_

Round 1

Reviewer 1 Report

The manuscript entitled " Towards a synthesis of the non-genetic and genetic views of cancer in understanding pancreatic adenocarcinoma initiation and prevention " was reviewed. This study mentioned that in addition to the genetic sources of carcinogenic mutations, transcriptional heterogeneity also plays an important role in the development of tumors. Through the analysis of cell subpopulations, the author believes that " edge sub-population " contribute to the carcinogenic transformation and are related to aging and microenvironment changes. However, the determination of this "marginal state" is not well defined and the relevance in tumorigenesis and other carcinogenic factors needs to be further understood.

Author Response

Lines 125-130 clarify the definition of the ‘edge state’ (which the reviewer refers to as the “marginal state”), which is that these are cells that do not express genes characteristic of conventional acinar cells and are transcriptionally much closer to a malignant cell than the other acinar cell. Lines 124-138, 169-173 and 184-185 refer to our published work in Cancer Research in 2021 (https://pubmed.ncbi.nlm.nih.gov/34049974/) where we establish the physiological and oncogenic relevance of the edge state. 

Briefly, in that work we find that the genes up-regulated in edge acinar cells in mice are highly similar to those that are up-regulated in KrasG12D-bearing acinar cells (Spearman Rho = 0.88 in our published work), suggesting that the tissue-specificity of Kras-G12D in driving PDAC may be because of pre-existing transcriptional variation in the Ras pathway in the mouse pancreas. We also show in our earlier work that edge-related genes are up-regulated during inflammation resolution and that edge cells are more frequent in older tissue donors. Combined with the epidemiological observation that a single bout of pancreas injury doubles the risk of PDAC upto ten years after injury resolution, we speculate that the edge state is very likely tied to PDAC initiation. The lack of availability of RNA-seq data from pancreatitis patients, and the late detection of PDAC makes it hard for us to causally link our edge cell signature to PDAC initiation. We hope that our review is able to spur efforts in data collection to establish a PDAC-priming transcriptional signature.

Reviewer 2 Report

This article provides perspectives of the involvement of non-genetic causes in the development of pancreatic adenocarcinoma. They did not deny the involvement of genetic factor KRAS(G12D) in this case but emphasized the importance of transcriptional and phenotypic heterogeneity cause in the development of pancreatic adenocarcinoma. And their argument was well-supported by the references they mentioned in the text. I think this article can be published in Cancers.

Minor point:

-Their sentences are rather long and complex, thus, sometimes, it is not easy to follow their arguments. It is recommended to use more simple and short sentences to help the audience to follow and understand their argument and points they want to deliver in this article.

-The abbreviation, PDAC and the full name, pancreatic adenocarcinoma does not match.

-The format of references (titles) needs to be checked if they followed the guideline of the Journal.

Author Response

This article provides perspectives of the involvement of non-genetic causes in the development of pancreatic adenocarcinoma. They did not deny the involvement of genetic factor KRAS(G12D) in this case but emphasized the importance of transcriptional and phenotypic heterogeneity cause in the development of pancreatic adenocarcinoma. And their argument was well-supported by the references they mentioned in the text. I think this article can be published in Cancers.

We thank the reviewer for their support of our work.

Minor point:

-Their sentences are rather long and complex, thus, sometimes, it is not easy to follow their arguments. It is recommended to use more simple and short sentences to help the audience to follow and understand their argument and points they want to deliver in this article.

We have shortened some long sentences and underlined those changes. 

-The abbreviation, PDAC and the full name, pancreatic adenocarcinoma does not match.

We have corrected the expansion of PDAC to pancreatic ductal adenocarcinoma (instead of pancreatic adenocarcinoma) in the title and lines 12, 22 and 110 accordingly.

-The format of references (titles) needs to be checked if they followed the guideline of the Journal.

Reviewer 3 Report

This article describes a novel concept of edge cells in PDAC. The article is well written. I enjoyed reading it. One minor suggestion: the purple colored cells in figure B should be annotated.

Author Response

We have now specified in the figure caption that the purple colored cells refer to malignant cells.